# *Plasmodium falciparum* parasite prevalence in East Africa: Updating data for malaria stratification

**Victor A. Alegana**[1,2]*, **Peter M. Macharia**[1,3], **Samuel Muchiri**[1], **Eda Mumo**[1], **Elvis Oyugi**[4], **Alice Kamau**[1], **Frank Chacky**[5], **Sumaiyya Thawer**[5,6,7], **Fabrizio Molteni**[5,6,7], **Damian Rutazanna**[8], **Catherine Maiteki-Sebuguzi**[8,9], **Samuel Gonahasa**[9], **Abdisalan M. Noor**[10], **Robert W. Snow**[1,10]

1 Population Health Unit, Kenya Medical Research Institute-Wellcome Trust Research Programme, Nairobi, Kenya, 2 Geography and Environmental Science, University of Southampton, Southampton, United Kingdom, 3 Centre for Health Informatics, Computing, and Statistics, Lancaster Medical School, Lancaster University, Lancaster, United Kingdom, 4 Division of National Malaria Programme, Ministry of Health, Nairobi, Kenya, 5 National Malaria Control Programme, Ministry of Health, Community Development, Gender, Elderly and Children, Dodoma, Tanzania, 6 Swiss Tropical and Public Health Institute, Basel, Switzerland, 7 University of Basel, Basel, Switzerland, 8 National Malaria Control Division, Ministry of Health, Kampala, Uganda, 9 Infectious Diseases Research Collaboration, Kampala, Uganda, 10 Centre for Tropical Medicine and Global Health, Nuffield Department of Medicine, University of Oxford, Oxford, United Kingdom

* Valegana@kemri-wellcome.org

**Data Availability Statement:** As part of a continued commitment by the KEMRI-Wellcome Trust's INFORM programme (http://inform-

## Abstract

The High Burden High Impact (HBHI) strategy for malaria encourages countries to use multiple sources of available data to define the sub-national vulnerabilities to malaria risk, including parasite prevalence. Here, a modelled estimate of *Plasmodium falciparum* from an updated assembly of community parasite survey data in Kenya, mainland Tanzania, and Uganda is presented and used to provide a more contemporary understanding of the sub-national malaria prevalence stratification across the sub-region for 2019. Malaria prevalence data from surveys undertaken between January 2010 and June 2020 were assembled form each of the three countries. Bayesian spatiotemporal model-based approaches were used to interpolate space-time data at fine spatial resolution adjusting for population, environmental and ecological covariates across the three countries. A total of 18,940 time-space age-standardised and microscopy-converted surveys were assembled of which 14,170 (74.8%) were identified after 2017. The estimated national population-adjusted posterior mean parasite prevalence was 4.7% (95% Bayesian Credible Interval 2.6–36.9) in Kenya, 10.6% (3.4–39.2) in mainland Tanzania, and 9.5% (4.0–48.3) in Uganda. In 2019, more than 12.7 million people resided in communities where parasite prevalence was predicted ≥ 30%, including 6.4%, 12.1% and 6.3% of Kenya, mainland Tanzania and Uganda populations, respectively. Conversely, areas that supported very low parasite prevalence (<1%) were inhabited by approximately 46.2 million people across the sub-region, or 52.2%, 26.7% and 10.4% of Kenya, mainland Tanzania and Uganda populations, respectively. In conclusion, parasite prevalence represents one of several data metrics for disease stratification at national and sub-national levels. To increase the use of this metric for decision making, there is a need to integrate other data layers on mortality related to malaria, malaria vector composition,

malaria.org/) to support sub-regional National Malaria Control Programs, the final geo-coded prevalence databases have been provided to the respective Ministries of Health to be integrated into national malaria data repositories. Data sharing and access is the responsibility of national governments and data requests can be directed to NMCPs. Kenya (EO – eloyugi@gmail.com and http://www.nmcp.or.ke/index.php/contact-us); Tanzania (FC – chackyfa@gmail.com and https://www.moh.go.tz/en/contact-us); and Uganda (DR – damianamanya@gmail.com and https://www.health.go.ug/contact-us). Kenya (http://www.nmcp.or.ke/index.php/contact-us); Tanzania (https://www.moh.go.tz/en/contact-us); and Uganda (https://www.health.go.ug/contact-us/). In 2017, data assembled since 1900 across Africa was released in the public domain and an updated data release of all additional data will be released in 2025.

**Funding:** VAA is funded by the Wellcome Trust Training Fellow (# 211208). PMM and AK are supported by the DELTAS Africa Initiative (DEL-15-003). The DELTAS Africa Initiative is an independent funding scheme of the African Academy of Sciences (AAS)'s Alliance for Accelerating Excellence in Science in Africa and supported by the New Partnership for Africa's Development Planning and Coordinating Agency with funding from the Wellcome Trust (# 107769) and the UK government and PMM acknowledges the support of the Royal Society under the Newton International Fellowship Scheme (NIF/R1/201418). RWS is supported as a Wellcome Trust Principal Fellow (# 212176) that also provided support to PMM, AK and EA. VAA, PMM, SM, EM, and RWS are grateful to the support of the Wellcome Trust to the Kenya Major Overseas Programme (# 203077). The authors also acknowledge support provided UK's Department for International Development for their continued support to a project Strengthening the Use of Data for Malaria Decision Making in Africa (DFID Programme Code 203155) that enabled the INFORM project to support National Malaria Control programmes in the region. The funders had no role in the study design, data collection and analysis, decision to publish, or preparation of the manuscript.

**Competing interests:** All authors declare no competing interests

**Abbreviations:** BCI, Bayesian Credible Interval; DHS, Demographic and Health Surveys; DIC, Deviance Information Criterion; EA, Enumeration Area; EVI, Enhanced Vegetation Index; GF, Gaussian Fields; GMP, Global Malaria Programme; GPS, Global Positioning Systems; GMRF, Gaussian

insecticide resistance and bionomic, malaria care-seeking behaviour and current levels of unmet need of malaria interventions.

## Introduction

The High Burden High Impact (HBHI) strategy of the World Health Organisations (WHO) Global Malaria Programme (GMP) encourages countries to use multiple sources of available data to provide a platform to understand the sub-national malaria risk. These national epidemiological frameworks should be used to rationalise finite resources and maximise the impact of malaria control [1, 2]. Ideally, data should be layered to provide meaningful stratifications based on the epidemiology of malaria risk and burden, areas of poor intervention coverage, vulnerability, and marginalisation to ensure health equity. Importantly, information must be resolved to administrative areas used by National Malaria Control Programmes (NMCPs) to guide sub-national stratification, policies, and resource allocation [2–4].

Varied data sources are increasingly used by countries across sub-Saharan Africa to measure the quantity of malaria risk at fine spatial resolution and decentralised health units. These include routine health system data on malaria test-positivity, case incidence, and screening for malaria infection among women attending ante-natal clinics [5]. Less common has been the sub-national variability in the composition of dominant vectors, their sibling species and respective Entomological Inoculation Rates (EIR) [6]. Historically, for over 100 years community-based surveys of malaria parasite prevalence have formed an important measure of malaria risk [7, 8]. Recently, these data have been collected from national household surveys undertaken every 3–5 years such as the Demographic and Health Surveys (DHS) or Malaria Indicator Surveys (MIS). However, because they lack adequate power for effective sub-national stratification on their own [9], they have been used in combination with other more opportunistic, assembled research survey data, sub-national monitoring data and school surveys, increasing the power of national data to provide estimates of risk at fine spatial scale. These data are often analysed using model-based geostatistical (MBG) methods [10] to predict risk in areas without data augmented by environmental geospatial covariates of transmission [11]. MBG incorporates measures of uncertainty of disease predictions at the population level. Modelled national parasite prevalence predictions have been used to inform sub-national approaches to malaria control in Kenya [12, 13], Uganda [14], Somalia [15, 16], Namibia [17], Senegal [18], Cote D'Ivoire [19, 20], Malawi [21–23], Angola [24], Madagascar [25], Ghana [26], Rwanda [27, 28], Mozambique [29, 30], Burkina Faso [31], Sudan [32] and Tanzania [33–35]. The use of routine health facility data sources to define malaria incidence has increased in recent years, in combination with parasite prevalence or independently, in Namibia [36], Zambia [37], Malawi [38], Tanzania [34, 35, 39], Madagascar [40–42], Zimbabwe [43], Ghana [44], Burkina Faso [45] and Uganda [46, 47].

Kenya has adopted a stratified, sub-national control response since 2010 [48], while Uganda [49] and Tanzania [50] have begun to use epidemiological data for stratified control following recommendations of the WHO's HBHI [2]. However, there is no sub-regional epidemiological stratification to tailor malaria control responses.

While community-based parasite prevalence data represent one layer in the sub-national stratification process, they are only as valuable as the quantity of data available spatiotemporally to compute contemporary predictions. Without updated empirical data, predictions are based largely on the relationship between geospatial covariates and historical data. The process

Markov Random Fields; HBHI, High Burden High Impact; INFORM, Information for Malaria; IPTp, Intermittent Presumptive Treatment Among Pregnant Women; MBG, Model Based Geo-Statistics; MCMC, Markov Chain Monte Carlo; MIS, Malaria Indicator Surveys; NEP, Non-exceedance probability; NMCP, National Malaria Control Programmes; NTL, Night-Time Light; $Pf$PR$_{2-10}$, age standardised parasite prevalence 2–10 years; PA$Pf$PR2-10, Population adjusted parasite prevalence 2–10 years; RDT, Rapid Diagnostic Test; R-INLA, Integrated Nested Laplace Approximations in R; RMSE, Root Mean Square Error; SPDE, Stochastic Partial Differential Equations; SSA, sub-Saharan Africa; TSI, Temperature Suitability Index; WHO, World Health Organization.

of stratification is dynamic, requiring constant revision, and integration into policy. As new data become available, sub-national stratification should be updated. This paper presents an updated assembly of community parasite survey data in Kenya, mainland Tanzania and Uganda using MBG inference to provide a more contemporary predictions of the sub-national malaria prevalence stratification across the sub-region for 2019.

## Methods

### Geographic scope and context

Kenya, mainland Tanzania and Uganda share national borders and since the launch of the Roll Back Malaria initiative in 2000, implemented simultaneously national malaria policies changes on case management and vector control. During the 1990s national parasite prevalence mapping efforts began in Kenya [51] and was repeated several times [12, 13]. Similar approaches were undertaken in Tanzania [34, 35] and Uganda [52]. In addition, the Information for Malaria (INFORM) project provided support to NMCPs in each country since 2015 in malaria risk mapping for decision making, feeding into National Malaria Strategic Plans (NMSPs) and applications for Global Fund support [53]. Each modelled product of parasite prevalence used community parasite prevalence data from different periods, different methods of spatiotemporal modelling and different spatial resolutions of prediction. The most recent parasite survey data used in developing prevalence maps in Kenya [12, 54, 55], Tanzania [34, 35, 50, 56] and Uganda [49, 57] were assembled between the 1990s and 2017. Each country recognises the changing heterogeneity of malaria transmission within its national borders and with time NMSPs have used the varying epidemiology to encompass a stratified, sub-national control response in Kenya [48, 55] and Tanzania [50].

### Parasite survey data assembly

Malaria parasite rate (PR) data from each of the three countries were assembled from existing data resources published in 2017 starting from January 2010 [58] and updated using published and survey data from 2017–2020. Data that precede 2010 are important historical information but are less valuable when informing the contemporary intensity of malaria transmission under the current levels of vector and parasite control. The processes of identifying, geo-coding, and standardizing community and school-based parasite survey data were described in detail elsewhere [8]. In brief, data search strategies included traditional peer-reviewed publication using PubMed, Google Scholar and Scopus using the free text keywords "*malaria*" and "*country-name*" and routine malaria publication alerts from Malaria World (http://www.malaria-world.com/). Importantly, the presence of the almost 30-year presence of the KEMRI-Wellcome Trust Programme in the sub-region has fostered a network of malaria scientists and ministry of health collaborators across the sub-region. These personal and institutional connections formed a significant part of the data search to identify higher spatial resolution of information in published and unpublished reports. All those who have contributed to this exercise are acknowledged for data provision for surveys undertaken between 2010 and 2020 in the (S1 Text).

Data comprised of the national survey of malaria infection in school children which started in Kenya in 2009 [59] and Tanzania in 2014 [60], subsequently repeated in Tanzania in 2017 and 2019 and Western Kenya in 2014 and 2019; the national household sample surveys of malaria indicators undertaken as part of DHS or MIS, where malaria infection was documented in children under 5 years (mainland Tanzania and Uganda) or under 15 years (Kenya); the multi-district mini-MIS to evaluate the impact of insecticide-treated nets in Uganda (2017–2019) [61] or baseline RTSS vaccine impact implementation studies (2019)

(CDC Kenya); and the continuous annual repeat surveys undertaken in specific sentinel demographic and epidemiological surveillance sites at Kilifi, Kakamega and Siaya (Kenya), Muheza (Tanzania) and Tororo, Jinja and Kanungu (Uganda). Data were reconciled to the smallest possible spatial extent covering a village or community, census enumeration area (EA) or school. Most data were within a 5 $km^2$ radius but were excluded if the spatial coverage exceeded >10 $km^2$.

The first phase of data assembled between 2010 and 2017 comprised of 4,771 space-time surveys in Kenya, Tanzania and Uganda and were published as open access in 2017 [8, 58]. This analysis includes new data assembled post-2017 to June 2020, resulting in the identification of new national household, school and research survey data. Furthermore, since the Open Access data release in 2017, the spatial and temporal consistency has been improved. Geographic coordinates of the individual-level survey and national household survey data were revised using village coordinates from the national statistical agency geo-coded village or EA databases, updated school databases, population settlement shapefiles from the Ministries of Lands, Google Maps and research surveys geolocated using Global Positioning Systems (GPS). For example, where information was available, it was possible to assign school children to their village of residence rather than aggregated to the school and continuous survey data from specific health and demographic surveillance sites. In addition, DHS and MIS data were matched to the EA and the centroid calculated. This enabled the identification of repeatedly sampled clusters over time as part of national household survey rounds and matched to surveys undertaken within sub-national areas by research groups. If a survey location was sampled by more than one group within a three-month time window the largest sample was selected. Where more than one sample cluster was sampled during the same survey within a single village or EA, these were combined as one time-survey location. This revised approach to the spatial aggregation of individual-level data improved the temporal and spatial congruence of survey sampling locations across the three countries.

Data extracted from each survey report included the name of the study location, survey dates (month and year), the upper and lower age of those included in the school/village/cluster sample, the number examined for malaria infection, the number positive, and the methods used to assess malaria infection (microscopy, Rapid Diagnostic Tests (RDTs) or combinations). Where multiple methods of diagnosis had been used, preference was given to microscopy where these were undertaken by research groups, or quality assured laboratories during national school/household surveys.

Given that there was a diversity in the age ranges of sampled populations between studies, it was necessary to standardise the variation in sampled ages to allow for comparison of *Plasmodium falciparum* parasite prevalence rate (*Pf*PR) in space and time. Algorithms based on catalytic conversion models were used to standardize the ages for each survey to 2–10 years assuming a modelled relationship between the upper and lower age of the sampled population, henceforth referred to as $PfPR_{2\text{-}10}$ [62, 63]. Parasite prevalence in children aged 2–10 years ($PfPR_{2\text{-}10}$) is predictive of other parameters of malaria transmission intensity such as the Entomological Inoculation Rate (EIR) and the Basic Reproduction Rate of Infection ($R_0$) [64]. As such values of $PfPR_{2\text{-}10}$ are widely used to model malaria control transmission reduction and the appropriate combinations of available interventions. They have also been used as a variable to predict probability of elimination [65]. Additionally, surveys varied in diagnostic methods to detect infection; either the visual presence of parasites in red cells or antigenaemia resulting from recent infections [66]. The aim was to standardise to microscopy and therefore RDT-only derived *Pf*PR measurements were converted to microscopy using a functional Bayesian *probit* regression relationship between RDT and microscopy [67].

## Covariates associated with $PfPR_{2-10}$

Covariates aid in predicting infection prevalence at fine spatial scales [68, 69], but can also be used for explanatory modelling [70, 71]. For predictive modelling, the accuracy is guided by the quantity of the empirical data, the measurement error in the covariates and model specification. Some previous approaches have excluded covariates due to concerns related to the mis-specification of the regression relationship between covariates and $PfPR_{2-10}$, resulting in invalid inferences in areas where empirical data are sparse or absent leading to $PfPR_{2-10}$ that is entirely driven by the covariates [12, 16, 34]. The inclusion of covariates can reduce the measure of prediction error without distorting the fidelity of regression relationships [71]. Therefore a parsimonious, minimal set of biologically plausible covariates for predictive modelling of $PfPR_{2-10}$ were selected from candidate lists of covariates guided by previous malaria infection prevalence modelling in Africa [69, 72, 73]. These included: precipitation, the Enhanced Vegetation Index (EVI), Temperature Suitability Index (TSI), night-time light (NTL), and aridity (S2 Text). TSI has also been used independently to define areas where low ambient temperature does not allow the parasite to survive long enough in the mosquito vector to support transmission (unsuitable/non-receptive/malaria-free areas) [74].

The age-adjusted and microscopy-adjusted survey data series were matched to the long-term climatic and ecological covariates and a selection procedure was implemented to select a parsimonious set of covariates that were important predictors of prevalence. This was computed using non-spatial generalised linear regression models implemented in the *bestglm* package in R [75] (S2 Text). Precipitation, TSI and NTL proved to be the set of covariates with the highest independent predictive power for $PfPR_{2-10}$ (S2 Text).

## Bayesian hierarchical geostatistical model for $PfPR_{210}$

Bayesian MBG approaches were used to interpolate survey data points at known space-time locations to provide posterior predictions based on parsimonious set of covariates for each period at unsampled locations with associated uncertainty. Since the three East Africa countries share national boundaries, the space-time data analysis was conducted jointly across the three countries using age-adjusted, microscopy-converted data collected between 2010 and 2020. The Bayesian hierarchical space-time model was implemented through a stochastic partial differential equations (SPDE) approach and using the Integrated Nested Laplace Approximations in R (R-INLA) [76, 77], to produce continuous maps of $PfPR_{2-10}$ for the year 2019. A Binomial likelihood was used with spatial effects introducing a measure of spatial autocorrelation. The spatial and temporal interaction was implemented using a space-time separable Matérn covariance function. The SPDE approach has a computation advantages over traditional Markov Chain Monte Carlo (MCMC) as the continuous domain Gaussian Field (GF) is represented as Gaussian-Markov Random Fields (GMRF) [78–80]. For complete hierarchical Bayesian model specification, more recent developments in R-INLA SPDE methodology allow for the selection of priors through model spatial range (the spatial distance at which the spatial correlation is small, often less than 0.01) and the field standard deviation [81, 82]. The selection of other prior information of the parameters (intercept, covariate distribution and residual error) followed the standard fixed prior specifications [68, 83]. Computation used the full spatial and temporal range of the data and aimed at estimating the continuous posterior mean of $PfPR_{2-10}$ and the 95% Bayesian credible intervals at $1 \times 1$ km spatial resolutions for 2019. Details of the Bayesian model specification are provided in S3 Text.

## Model validation

A 20% spatially and temporally representative validation set of $PfPR_{2-10}$ survey data were selected randomly based on equal selection probability. Prediction statistics computed from

the validation set included the mean square error, the correlation between the predicted and observed vector densities, and the root mean square error (RMSE). The correlation coefficient provides a simple measure of linear association between the data and prediction sets, and RMSE is a measure of the average accuracy of individual predictions.

### Prediction resolutions adjusted for population density

Population and malaria risk are not evenly, or congruently, distributed in space. Thus, population density and $PfPR_{2-10}$ were combined to provide a population-weighted parasite prevalence (PA$PfPR_{2-10}$). Pixels, where a zero value of TSI was regarded as zero probability of infection and zero population would not be included in risk attribution. The population density was assembled from the WorldPop gridded data that uses random-forest modelled disaggregation of latest census data projected to 2019 [84–87]. The gridded population data are available for at 100 m x 100 m spatial resolution and raster data were resampled to a matching grid of 1 x 1 km corresponding to the $PfPR_{2-10}$ surface.

The predicted PA$PfPR_{2-10}$ was classified into 6 endemicity strata corresponding to classes used currently in Tanzania [50]: areas which were unsuitable for transmission (malaria-free, TSI = 0), areas of very low risk where PA$PfPR_{2-10}$ <1%; low risk 1% to <5% (low-risk strata 1); 5% to <10% (low-risk strata 2); moderate risk PA$PfPR_{2-10}$ 10% to <30%; and the high endemicity areas of PA$PfPR_{2-10}$ ≥30% (high risk). The samples from posterior predictive distributions, weighted for the population were aggregated to the 2019 spatial units representing health decision-making units used for sub-national stratification: 47 counties in Kenya, 184 councils in Tanzania, and 135 districts in Uganda. Within each administrative polygon, the aggregated means were summed to provide a risk class for each health unit using strata defined earlier.

### Ethics statement

The study involved the assembly of aggregated secondary data, previously published or part of national anonymised surveys. Ethical approvals for all specific survey data assembled was presumed sought by national investigators.

## Results

### Survey data assembly

Since the release of survey data from Kenya, Tanzania and Uganda in 2017, a total of 14,170 (74.8%) new time-space survey data were identified. One data point was excluded because it was not possible to find a reasonable geo-position, one survey location was excluded because it represented an area >10 km$^2$. The final age-standardised and microscopy-converted data represented 18,940 time-space surveys between January 2010 and June 2020 at 6,866 unique spatial locations (Fig 1; Fig 2A). 7122, 4342 and 7476 surveys were from Kenya, mainland Tanzania and Uganda respectively (Fig 2A). 11,642 (61.5%) surveys were undertaken among samples ≥10 individuals; 4729 (25.0%) were a result of school surveys, 2366 (12.5%) from MIS, 4259 (22.5%) among sub-national MIS, and 682 (3.6%) from malaria modules within DHS (Fig 2B).

### Malaria parasite prevalence in 2019

Based on the 20% subset data, the mean error in the prediction of $PfPR_{2-10}$ for 2019 for the covariate-adjusted model revealed low bias with a slight tendency to underpredict by -0.007 (mean prediction error). The average error associated with predictions (absolute error) was

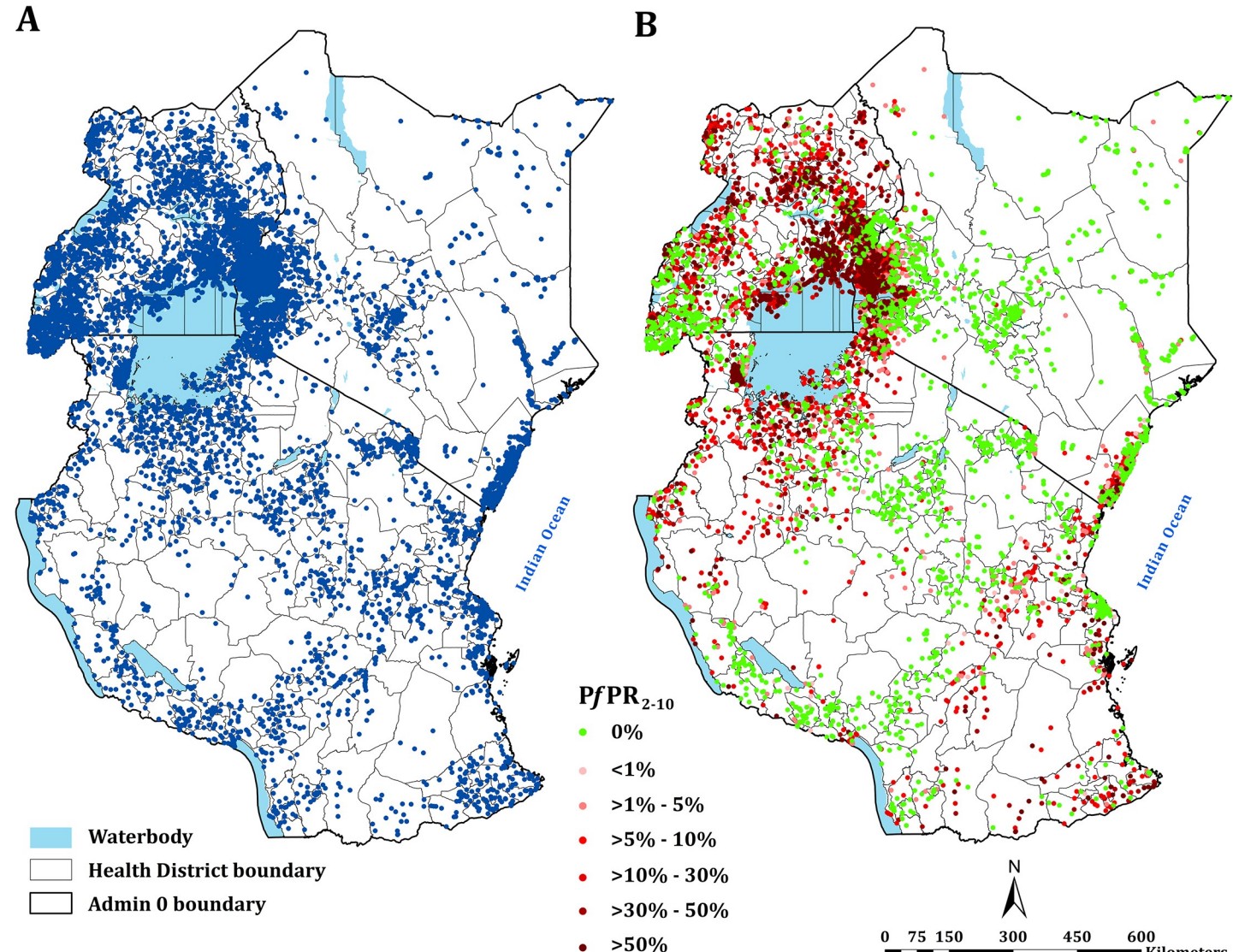

**Fig 1. Assembled parasite rate surveys. (A)** Distribution of all assembled survey data ($n$ = 18940) between 2010–2020; **(B)** the distribution of age-corrected and microscopy-standard parasite prevalence ($Pf$PR$_{2-10}$) estimates among samples ≥10 individuals with the highest values on top when multiple surveys conducted at the same location. Base shapefiles used in all figures downloaded from: Kenya– https://data.humdata.org/dataset/ken-administrative-boundaries; Uganda– https://data.humdata.org/dataset/uganda-administrative-boundaries-admin-1-admin-3 and Tanzania–https://data.humdata.org/dataset/tanzania-administrative-boundaries-level-1-to-3-regions-districts-and-wards-with-2012-population https://gadm.org/.

0.02 suggesting a good model precision. The correlation between the actual and predicted values for the hold-out set was 0.88 indicating a strong linear agreement between observed values and predictions.

The continuous maps of PA$Pf$PR$_{2-10}$ for 2019 is shown in Fig 3A and by each of the six endemicity strata in Fig 3B. At a national level, the PA$Pf$PR$_{2-10}$ was 4.7% (95% Bayesian Credible Interval 2.6–36.9) in Kenya, 10.6% (3.4–39.2) in Tanzania, and 9.5% (4.0–48.3) in Uganda. For all three countries, spatial heterogeneity in parasite prevalence is evident, ranging from unsuitable for transmission to the highest predictions above 77% (Fig 3A). Malaria prevalence was higher around the Lake Victoria basin, North-Western Tanzania, Northern Uganda and the southern coastal tip of Kenya (Fig 3A). A large swathe of Kenya and Tanzania running

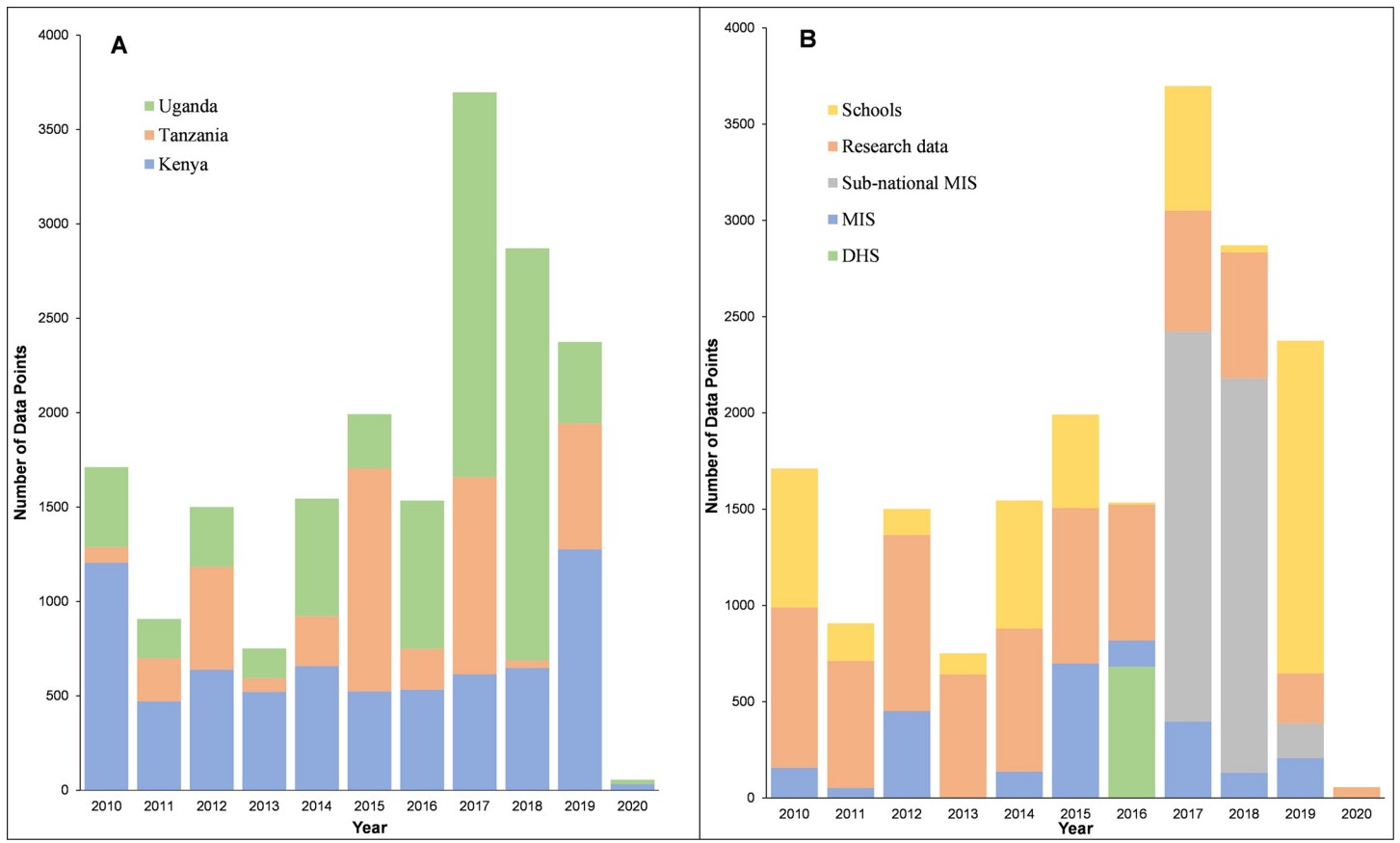

**Fig 2. Assembled surveys by year and data source. (A)** Temporal distribution of surveys 2010–2020 by country **(B)** Temporal distribution of surveys 2010–2020 according to the data source

from North Eastern Kenya to Lake Malawi in southern Tanzania predicted to have very low malaria transmission (Fig 3A and 3B). In 2019, more than 12.7 million people resided in communities where parasite prevalence was predicted to be ≥ 30%, including 6.4%, 12.1% and 6.3% of Kenya, mainland Tanzania and Uganda population, respectively (Fig 3B; Table 1). Conversely, areas that supported very low parasite prevalence (<1%) but were suitable for transmission (TSI >0), consisted of approximately 46.2 million people across the sub-region, or 52.2%, 26.7% and 10.4% of Kenya, mainland Tanzania and Uganda populations respectively (Fig 3B; Table 1).

The mean PA$P\!f$PR$_{2\text{-}10}$ per decentralised health administrative unit in Kenya (47 counties), mainland Tanzania (184 councils) and Uganda (135 districts) are shown in Fig 4, representing the class of parasite prevalence predicted for each unit in 2019 (S1 Table). Overall, 25 health administration units had mean PA$P\!f$PR$_{2\text{-}10}$ ≥30%, these high transmission settings were uncommon across the entire sub-region in 2019: two counties in Kenya (Busia and Siaya), seven districts in Uganda (Apac, Busia, Namayingo, Luuka, Jinja, Oyam and Kwania); and sixteen councils in mainland Tanzania (Chato, Biharamulo, Bukombe, Nanyamba, Tarime, Ushetu, Kasulu, Buchosa, Tarime TC, Geita TC, Ukerewe, Geita, Tandahimba, Mbogwe, Nyangwale and Mtwara Rural). None of the health administrative units could be classified as entirely unsuitable for malaria transmission. However, in Kenya Nairobi county was classified as having a mean PA$P\!f$PR$_{2\text{-}10}$ of 0.1% in 2019, this urban extent, with high levels of mobility could not be classified uniquely as very low transmission but was regarded as malaria-free

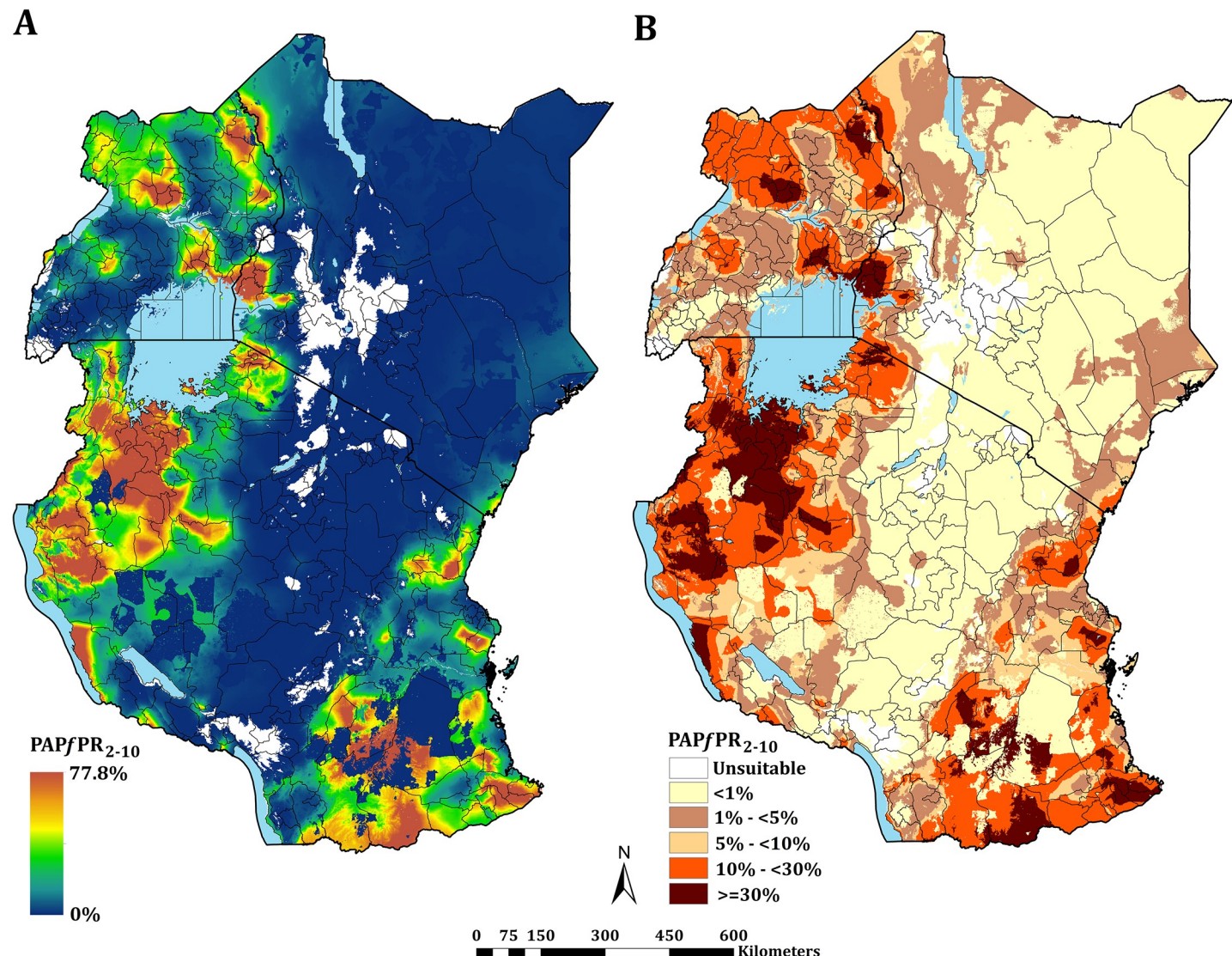

**Fig 3. Predicted mean PA*Pf*PR₂₋₁₀ at 1 × 1 km spatial resolution maps in 2019. (A)** mean prevalence (continuous stretched scale), and **(B)** Classified mean of the endemicity. The white represents the climatic unsuitability for transmission (TSI = 0). PA*Pf*PR₂₋₁₀ predictions are shown for areas within the stable limits of transmission.

[88]. 28 (59.6%), 51 (27.7%) and 17 (12.6%) health units in Kenya, mainland Tanzania and Uganda, respectively, were predicted to be of very low risk (PA*Pf*PR₂₋₁₀ < 1%) in 2019. For health policy planning purposes, it is notable that across all countries, 244 (66.7%) of the health administrative units were defined as <10% PA*Pf*PR₂₋₁₀ in 2019. The numbers of health units allocated to each of the five, suitable endemicity classifications, and populations in these averaged risk units are provided in Table 1.

## Discussion

Over 18,000 space-time empirical parasite survey data observations on community-based malaria prevalence were assembled from multiple sources since 2010. This represented an important data resource, where new information was identified since malaria prevalence maps were previously developed in Kenya, mainland Tanzania and Uganda. This large dataset

**Table 1. The estimated population at risk (percentage) for all-ages in each malaria endemicity class in 2019.**

| | The population at risk 2019 per endemicity class | | | | | |
|---|---|---|---|---|---|---|
| | Population from the Continuous surface (Fig 3B) | | | Number and Population in the Health units (Fig 4) | | |
| Endemicity Classes | Kenya | Tanzania | Uganda | Kenya (Counties) | Tanzania (Councils) | Uganda (Districts) |
| Malaria Free | 7,225,608 (14.0) | 1,837,735 (3.3) | 1,604,742 (3.7) | 0 (0.0) | 0 (0.0) | 0 (0.0) |
| | | | | 0 (0.0) | 0 (0.0) | 0 (0.0) |
| Very low (<1%) | 26,876,619 (52.2) | 14,788,527 (26.7) | 4,578,961 (10.4) | 28 (59.6) | 51 (27.7) | 17 (12.6) |
| | | | | 30,700,484 (59.6) | 13,978,579 (25.3) | 3,795,512 (8.7) |
| Low 1 (1%—<5%) | 7,019,389 (13.6) | 12,488,093 (22.6) | 16,068,838 (36.7) | 9 (19.1) | 29 (15.8) | 44 (32.6) |
| | | | | 9,463,322 (18.4) | 12,549,065 (22.7) | 17,384,567 (39.7) |
| Low 2 (5%—<10%) | 3,399,312 (6.6) | 7,383,339 (13.3) | 6,653,000 (15.2) | 3 (6.4) | 34 (18.5) | 29 (21.5) |
| | | | | 2,576,949 (5.0) | 8,780,939 (15.9) | 8,028,560 (18.3) |
| Moderate (10%—<30%) | 3,693,499 (7.2) | 12,130,029 (21.9) | 12,149,932 (27.7) | 5 (10.6) | 54 (29.3) | 38 (28.1) |
| | | | | 6,838,613 (13.3) | 14,542,473 (26.3) | 12,148,512 (27.7) |
| High (> = 30%) | 3,304,422 (6.4) | 6,689,688 (12.1) | 2,772,542 (6.3) | 2 (4.3) | 16 (8.7) | 7 (5.2) |
| | | | | 1,939,482 (3.8) | 5,466,361 (9.9) | 2,470,861 (5.6) |
| Total | 51,518,850 (100) | 55,317,472 (100) | 43,828,014 (100) | 47 (100) | 184 (100) | 135 (100) |
| | | | | 51,518,850 (100) | 55,317,417 (100) | 43,828,012 (100) |

reflects the importance of data sharing at national levels and harnesses considerable information beyond single national, periodic household surveys. Over 80% of the data included here were generously provided by research groups located in each country (S1 Text).

Combining data across the three countries allows information to be shared for modelling malaria risk across borders. Understanding cross-border risk become increasingly significant as countries aim for sub-national elimination [89]. Border communities represent an important migrant population, for example, nomadic pastoralist groups that transect the low transmission borders of Kenya and Tanzania (Fig 3A) and the communities that share high transmission in all three countries bordering Lake Victoria (Fig 3A).

Using the available data since 2010 within a time-space MBG framework predicted that only 8.5% of East Africa's population lived in areas that support high transmission (PA$Pf$PR$_{2-10}$ $\geq$ 30%) in 2019. This varied between Kenya (6.4%), Tanzania (12.1%) and Uganda (6.3%) (Fig 3A, 3B; Table 1). These high parasite prevalence areas were described during more recent parasite prevalence risk mapping [12, 34, 52] and covered 2 counties, 16 councils, 7 districts in Kenya, mainland Tanzania and Uganda, respectively (Fig 4; Table 1; S1 Table). These areas represent the most vulnerable populations to high disease burdens and intensive malaria control efforts are required to reduce transmission using available vector control, drug based-prevention strategies, and malaria vaccines once approved for wide-scale use. Despite decades of promoting universal LLIN and intermittent presumptive treatment among pregnant women (IPT$p$) coverage nationally, there is a need to re-design distribution that targets the higher prevalence areas. This was initiated in Kenya in 2010, restricting routine and mass campaigns to high-risk counties in Western and Coastal Kenya [48] and has become part of a revised national strategic planning in Tanzania [50].

The very low transmission belt represented 28 counties, 51 councils, and 17 districts in Kenya, Tanzania and Uganda, respectively (Fig 4; Table 1; S1 Table) covered a population of over 46.2 million people. In an average year, most children and pregnant women may not encounter infections and overall disease burdens will continue to be of low prevalence. New vaccines or presumptive use of drugs to reduce disease burdens in these areas would not be cost-effective. These areas of very low infection prevalence represent sub-national regions for

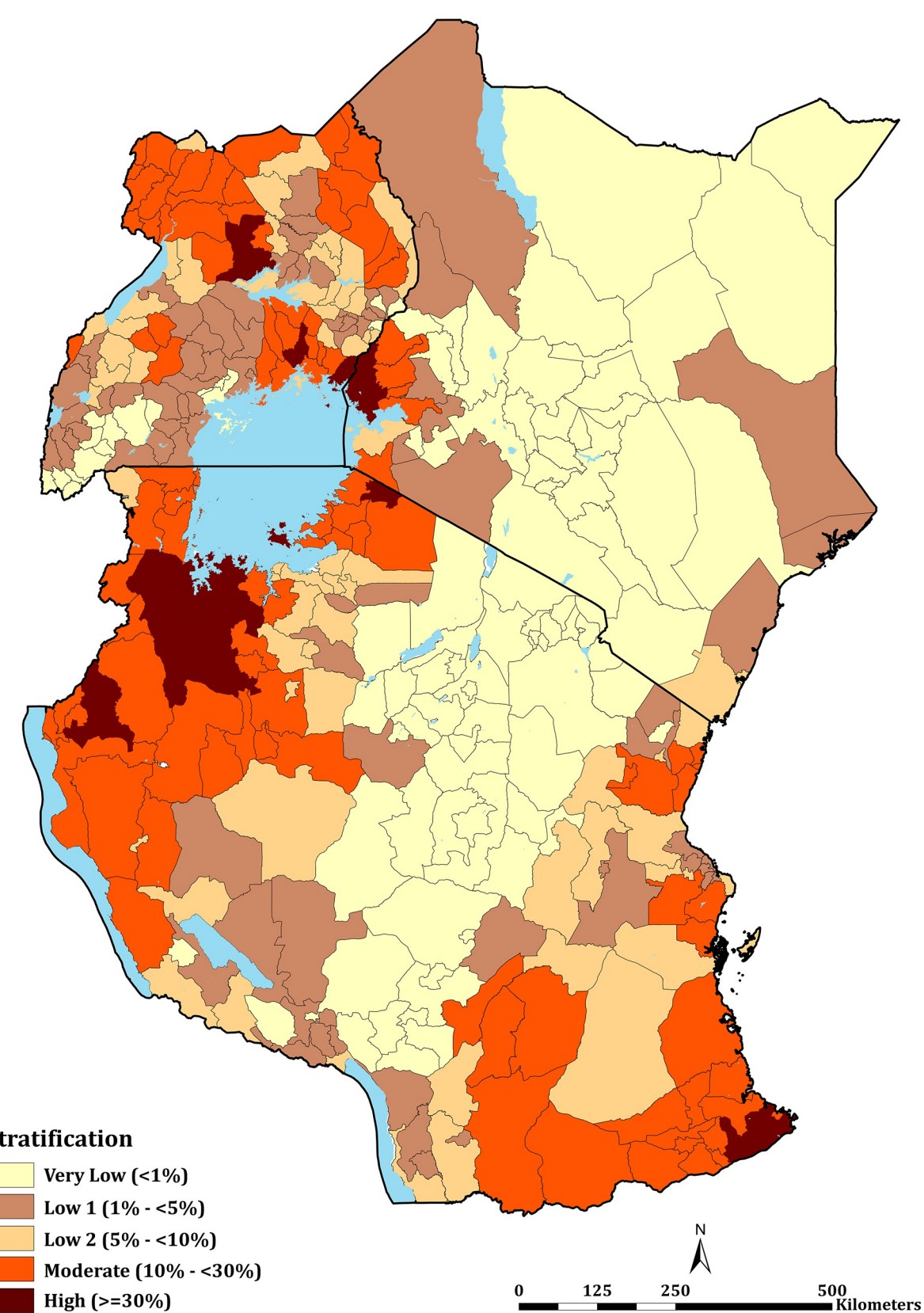

**Fig 4. East Africa *Pf*PR$_{2-10}$ stratification.** Stratification of health decision-making units based on the level of PA*Pf*PR$_{2-10}$ (aggregated mean) for 2020. These comprised 47 counties in Kenya, 184 councils in mainland Tanzania, and 135 districts in Uganda (see S1 Table).

pre-elimination [90, 91]. In Tanzania [50] and Kenya [55], the most recent national malaria strategies have identified these areas of each country as an immediate plan for a pathway to elimination. This will require additional resources to mount early case detection, foci detection, mapping risk at a fine spatial resolution, requiring a robust and reactive health information system [92]. However, currently, routine case detection in these countries is inadequate to serve as either an intervention or a means to characterise transmission in very low-risk areas. Recent data from Kenya [93] and Uganda [94] suggest a poor performance in the test-track-treat policy in areas of low-malaria risk compared to more stable, endemic areas, affecting both case management and reliability of routine data.

Considering sub-national elimination, however, requires confidence in the stratification predictions. The MBG approaches come with statistical uncertainty, which might arise from inadequate survey input data (suggesting further sampling needs) and/or inherent variability in small area prediction. In previous work in Kenya and Somalia [12, 16], a 90% certainty in the predictions of PA$Pf$PR$_{2-10}$ being <1% (Non-Exceedance Probability, NEP) have been used for statistical confidence in the MBG predictions for programmatic certainty. Applying this approach to the 2019 PA$Pf$PR$_{2-10}$ predictions across East Africa there is a high congruence with predicted and 90% NEP certainty (Fig 5). Among the 96 health administrative units predicted to be <1% in 2019, 70 had more than 90% of the population living in areas where there was a 90% certainty that PA$Pf$PR$_{2-10}$ was <1% (S1 Table and population map in S1 Fig).

In the absence of reliable routine data, current sampling strategies for parasite prevalence are poorly suited to characterise areas of very low parasite prevalence and require larger sampling effort to increase precision [95]. Both Microscopy and RDTs are routinely used to assess malaria infection but RDTs in particular could underestimate level of infection in low transmission setting [96]. Reducing the uncertainty and improving confidence in stratification of very low-risk areas should be encouraged and may require only one-off large-scale community and/or school surveys including PCR and serology [97] and could be restricted to areas of highest existing statistical uncertainty.

Importantly, areas that migrate to very low risk from low or moderate risk should be handled carefully in reverting immediately to a different mix of intervention and avoid risk of malaria rebounding, remaining cognisant of recent receptive risks [15, 17]. In Uganda, Apac district achieved a rapid reduction in parasite prevalence through indoor residual house-spraying, which after stopping resulted in an immediate rebound [98]. A more cautious use of PA$Pf$PR$_{2-10}$ < 1% to transition intervention is to remain in this class for a fixed number of years pre-transition to a different intervention-mix class, an enhanced case-detection is established, of proven fidelity and used to constantly review for a potential resurgence.

In between the two extremes of very low and high parasite prevalence people lived in predominantly low malaria prevalence areas (PA$Pf$PR$_{2-10}$ 1%-<10%; 52.8 million people) and areas of moderate transmission (PA$Pf$PR$_{2-10}$ 10%-<29%; 28.0 million people), covering 17, 117 and 90 health administrative units in Kenya, Tanzania and Uganda respectively (Fig 4; Table 1; S1 Table). Whether combinations of available interventions should be tailored across these 3 risk strata remains uncertain, except for maintaining prompt effective diagnosis, treatment, and referral, which must remain constant across all strata. It is important to recognise that the classification of strata based solely on parasite prevalence is arbitrary, there will be overlapping risks based on statistical confidence in the prediction [16]. A more conservative priority setting would be to include these counties, councils or districts in the accelerated, intensive efforts to reduce parasite burdens. Having multiple risk strata, however, allows a national control programme to review progress in moving decentralised health units down a continuum of risk, to set milestones for progress and constantly assess areas of intractable high prevalence and the possible resurgence of risk.

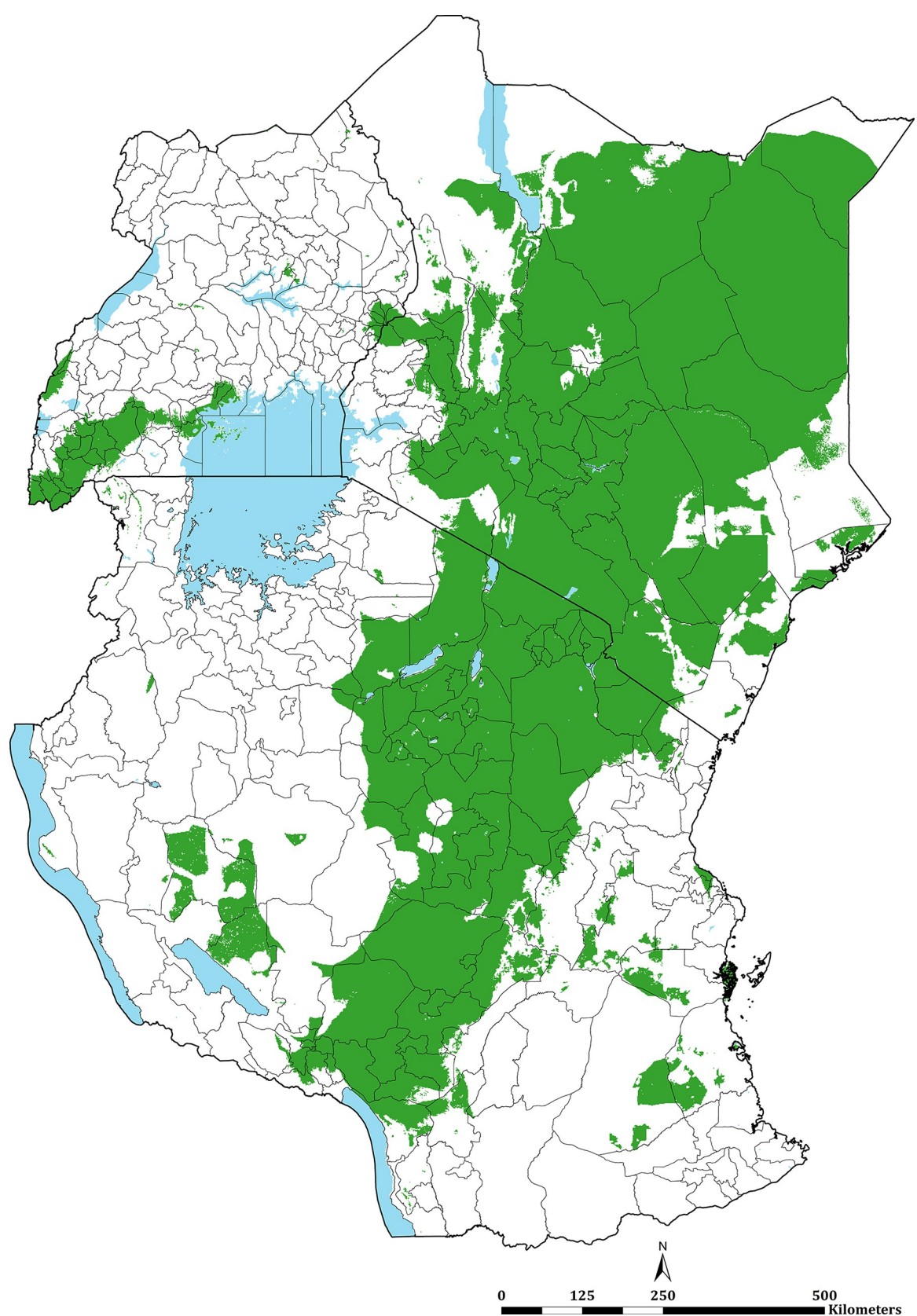

**Fig 5. Non-exceedance probability (NEP) maps for 2019.** PA*Pf*PR$_{2-10}$ predictions are 90% certain to be < 1%, shown in green. Derived from the fitted spatiotemporal model, formally expressed as: NEP = (Prob PA*Pf*PR$_{2-10}$ (x, t) < l|Data); where *l* is the prevalence

threshold. A NEP close to 100% indicates that PA$Pf$PR$_{2-10}$ is highly likely to be below the threshold $l$; if close to 0%, PA$Pf$PR$_{2-10}$, is highly likely to be above the threshold $l$; if close to 50%, PA$f$PR$_{2-10}$, is equally likely to be above or below the threshold $l$, hence corresponding to a high level of uncertainty.

The modelled data presented here included very little data post-2019 (Fig 2A). Predicting beyond 2019 would have been inappropriate in the absence of data. However, using parasite prevalence as one layer in malaria stratification is a dynamic process, repeated every 2–3 years. Data from the November 2020 MIS in Kenya was not available by June-2021 and nationwide school surveys are planned for 2022 in Kenya and mainland Tanzania accompanying a further MIS in 2021. In Uganda an MIS has been planned for 2023 and national research partners continue to maintain surveillance in specific areas of each country. The exercise of parasite prevalence data assembly and modelling would ideally be repeated in 2023. Future mapping can be improved by including more temporally varying covariate layers rather than static covariates such as TSI. Here only long-term means were considered to resolve risk to the year of prediction. Malaria risk could however vary inter and intra annually requiring additional sampling to characterise seasonality in parasite prevalence.

The use of parasite prevalence has both disadvantages and advantages as a component of risk stratification. Sampled community or school populations represents an infection prevalence estimate on one day. National household surveys (DHS, MIS) are not powered to provide sub-national, health unit precision, are expensive and data are often only available to the national malaria control programmes 12–24 months after completion owing to survey agency and national statistical office agreements, beyond the control of the NMCP. School surveys offer a much cheaper alternative to DHS/MIS approaches to sampling parasite prevalence [19, 59, 60] and NMCPs have more direct control over data collection and access. School surveys have become increasingly important in recent years, over 25% of the point estimates included in the present analysis came from school-based surveys (Fig 2B). Routine data are available 365 days a year, and far more powerful to estimate seasonal intra- and inter annual variations in malaria risk. However, they represent risks among those who seek care, depend on the diagnostic capacities of health facilities and pre-existing acquired immunity, as such these data are not directly comparable to the risks of infection among community members. Both are important, and more work is required in understanding the relationships between community infection and infection among those who seek care [99] and how these might be jointly modelled (hybrid mapping) rather than treated as separate entities [100, 101].

Parasite prevalence is only one metric in the stratification process. To increasing its utility in decision making, it should be combined with routine data alongside other aspects of population vulnerability including the possibilities for urban control. Other data have been less well used in stratification, for example layering information on decentralised, health unit estimations of malaria mortality burdens, vector composition, insecticide resistance and bionomics, access to curative services and unmet needs based on existing vector control coverage. There is a need to build data acquisition processes and capacities within national malaria control programmes and the statistical handling of this information within a multi-layered data platform to make more effective decisions on malaria control efficiencies. National data repositories, improved data sharing and increased use of data nationally is paramount. These data must be country-owned and their use country-driven [102, 103].

## Supporting information

**S1 Text. Data acknowledgements.**
(DOCX)

**S2 Text. Covariate selection.**
(DOCX)

**S3 Text. Spatiotemporal modelling of malaria risk.**
(DOCX)

**S1 Table. PA$Pf$PR$_{2\text{-}10}$ estimates.**
(DOCX)

**S1 Fig. Additional maps.**
(DOCX)

## Acknowledgments

The many institutions and individuals who have contributed data are acknowledged in Supplementary Information 1. In addition, the authors wish to acknowledge the additional technical help provided in data location, geocoding and assembling additional GIS surfaces at the KEMRI-Wellcome Trust programme including David Kyalo, Laurissa Suiyanka and Lydia Mwangi. We would like to thank national control programme managers in Kenya (George Githuku, Elvis Oyugi), Uganda (Jimmy Opiyo) and Tanzania (Ally Mohammed) for formative discussions over several years on malaria risk mapping in the sub-region and Philip Bejon for comments on earlier drafts.

## Author Contributions

**Conceptualization:** Victor A. Alegana, Peter M. Macharia, Robert W. Snow.

**Data curation:** Peter M. Macharia, Samuel Muchiri, Eda Mumo, Alice Kamau, Frank Chacky, Robert W. Snow.

**Formal analysis:** Victor A. Alegana.

**Funding acquisition:** Victor A. Alegana, Peter M. Macharia, Robert W. Snow.

**Investigation:** Victor A. Alegana.

**Methodology:** Victor A. Alegana, Peter M. Macharia, Samuel Muchiri, Robert W. Snow.

**Project administration:** Robert W. Snow.

**Resources:** Robert W. Snow.

**Software:** Victor A. Alegana.

**Supervision:** Victor A. Alegana, Robert W. Snow.

**Validation:** Victor A. Alegana, Robert W. Snow.

**Visualization:** Victor A. Alegana, Samuel Muchiri, Eda Mumo, Robert W. Snow.

**Writing – original draft:** Victor A. Alegana, Robert W. Snow.

**Writing – review & editing:** Victor A. Alegana, Peter M. Macharia, Samuel Muchiri, Eda Mumo, Elvis Oyugi, Alice Kamau, Frank Chacky, Sumaiyya Thawer, Fabrizio Molteni, Damian Rutazanna, Catherine Maiteki-Sebuguzi, Samuel Gonahasa, Abdisalan M. Noor, Robert W. Snow.

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
