## [Decision Letter · Decision Letter 0]

19 Aug 2021

 PGPH-D-21-00103

 Plasmodium falciparum parasite prevalence in East Africa: updating data for malaria stratification.

Dear Dr. Alegana,

Thank you for submitting your manuscript to PLOS Global Public Health. After careful consideration, we feel that it has merit but does not fully meet PLOS Global Public Health’s publication criteria as it currently stands. Therefore, we invite you to submit a revised version of the manuscript that addresses the points raised during the review process.

 There is agreement among the 3 reviewers that the statistical methods presented are sound, however some additional improvements and clarifications have been suggested. Please provide a response to each of the comments, and address the methodological clarifications that have been flagged. Regarding reviewer 2's comment on data availability, please confirm if the survey data since 2017 is expected to be included in your data repository, and the justification for release of this data being postponed until 2025. Note that the PLOS Data policy requires authors to make all data underlying the findings described in their manuscript fully available without restriction, with rare exception.

We look forward to receiving your revised manuscript.

Kind regards,

Ruth Ashton, Ph.D.

Academic Editor

Journal Requirements:

Additional Editor Comments (if provided):

Reviewers' comments:

Reviewer's Responses to Questions

**Comments to the Author**

1. Does this manuscript meet PLOS Global Public Health's publication criteria? Is the manuscript technically sound, and do the data support the conclusions? The manuscript must describe methodologically and ethically rigorous research with conclusions that are appropriately drawn based on the data presented.

Reviewer #1: Yes

2. Has the statistical analysis been performed appropriately and rigorously?

Reviewer #1: Yes

3. Have the authors made all data underlying the findings in their manuscript fully available (please refer to the Data Availability Statement at the start of the manuscript PDF file)?

Reviewer #1: No

4. Is the manuscript presented in an intelligible fashion and written in standard English?

Reviewer #1: Yes

5. Review Comments to the Author

Reviewer #1: Overall: This analysis involved a large data collection effort to produce an analysis of programmatic relevance. The outputs are clear and operationally relevant. Some methodological decisions could use some further justification and areas of the results and discussion could be expanded to complete the picture of how this analysis – and others like it to come – may be used by programs for strategy decisions.

Major comments:

1. Clarify the abstract

2. Methods – some additional justification for how the limited set of covariates were selected from the three references cited would be useful. There could be additional land cover, potential evaporation index, etc variables of interest. The covariates with the most positive predictive power were identified, but were methods explored to identify the dominant covariates in different operational units? That type of information would be useful for programs.

More information on how time was used – time of survey and matching to the time of covariates would be useful – were monthly covariates and/or lagged variables used?

3. Results – it would be useful to have more presentation of the certainty of the predictions in the results section. Perhaps the non-exceedance map here and an exceedance map as well.

4. Discussion – it would be interesting to have a bit more information about how these outputs could be used alongside the routine data that programs are familiar with and rely on. In spite of the biases of routine data – as you mention – there are many advantages to it like learning the seasonal signals of transmission.

Minor comments:

1. Abstract – in the introduction, it would be good to introduce that you are providing a modeled output and not simply presenting the PR data available (as you do at the end of the introduction). With your modeled outputs there are stratification values beyond the sampled areas.

In the methods section, the wording of data being “undertaken” is a bit confusing. Perhaps say something like, “Malaria prevalence data from surveys undertaken between January 2010 and June 2020 were assembled from each of the three countries”.

Do you adjust the data by the population, enviro, and eco covariates? Or did you simply use them as covariates in the modeling workflow?

Conclusion – I am curious how some of the additional data types listed would help increase the use of prevalence data? PR data seem independent of some of these (for example, care seeking). Would stratification be more useful of these additional metrics were accounted for alongside PR data?

2. Introduction – in the abstract you mention the Global Technical Strategy and here HBHI. Is the use of multiple metrics for planning quoted in both strategies?

What do you mean by vulnerability in the first paragraph? Like environmental receptivity? Or the impact of resource scarcity/vulnerability on different populations?

Paragraph 2 – are surveys not powered to support sub-national stratification? Or is that they lack complete geographic coverage? If it’s both – then those would be two separate issues worth mentioning.

Not sure ‘interpolation’ is the right to say what MBG was used for

Great to see the list of countries with country-specific prevalence maps. Does the first list use PR data?

3. Methods – the geographic scope and context section seems to belong in the introduction section.

Parasite data survey data assembly – it would be useful to have citations for what national PR survey sources are included in the existing data sources [ref 58]. Is that what paragraph 3 of this section outlines? Or we these included in addition from what is in reference 58? The first sentence of the third paragraph in this section seems like a list – are you saying that these are surveys that were gathered?

Great to list the updates and improvements made to the 2017 open access data. What resolution had the DHS and MIS data been at previously? Not the cluster level? The regional level?

For the study location – lat/longs were given for point observations I assume, but how were areal units like EAs treated? Was the centroid used? Or the whole area?

Were the survey data matched to covariates by month? Year? How were areal units matched?

Prediction resolutions adjusted for population density – was zero risk given to pixels with 0 TSI and 0 population or either or? Was there any consideration for comparing the WorldPop data to the CIESEN population data (https://data.humdata.org/dataset/highresolutionpopulationdensitymaps)?

4. Results – an impressive amount of new data assembled since the 2017 publication! Great to see. For Figure 1 – the reds in the colour scale used for B and C are hard to tell apart. Figure 2 is great. It would be useful to see – maybe in an SI – how the colors shown in Fig 2B would map spatially.

Figure 3 – are the very low PR areas in W Tanzania parks/game reserves? Is that an artifact of the population surface?

Figure 4 and Table 1 are great for programmatic use. Would be great to see what you have in SI 4 as a map as well.

5. Discussion – are the 18,000+ surveys assembled since 2010 for this region specifically?

What about limitations with the methods? Aside from the data only? And data limitations beyond the survey data? Is the TSI surface from 2011? Issues with the population data?

6. PLOS authors have the option to publish the peer review history of their article (what does this mean?). If published, this will include your full peer review and any attached files.

**Do you want your identity to be public for this peer review?** For information about this choice, including consent withdrawal, please see our Privacy Policy.

Reviewer #1: No

**Comments to the Author**

1. Does this manuscript meet PLOS Global Public Health’s publication criteria? Is the manuscript technically sound, and do the data support the conclusions? The manuscript must describe methodologically and ethically rigorous research with conclusions that are appropriately drawn based on the data presented.

Reviewer #2: Yes

Reviewer #3: Yes

2. Has the statistical analysis been performed appropriately and rigorously?

Reviewer #2: Yes

Reviewer #3: Yes

3. Have the authors made all data underlying the findings in their manuscript fully available (please refer to the Data Availability Statement at the start of the manuscript PDF file)?

Reviewer #2: No

Reviewer #3: Yes

4. Is the manuscript presented in an intelligible fashion and written in standard English?

Reviewer #2: No

Reviewer #3: Yes

5. Review Comments to the Author

Reviewer #2: The paper is essentially an update to the previous malaria prevalence maps published by the same group across the East Africa region. The overall goal of this work is to feed into the malaria risk stratification planning that has been highlighted by the WHO as critical piece to help with targeting intervention mixes appropriately and monitor the progress of malaria burden. This paper highlights that prevalence estimates are simply one of the key surfaces in the grand scheme that is stratification, but not the final outcome, and should be used appropriately in conjunction with other burden metrics that the countries collect. Whilst the methodology nor data may not be ground-breaking it is certainly an important piece of work operationally. I don’t have major comments besides one:

1. Whilst I commend that the authors have shared these data back to their respective ministries of health and agree that work should be country led, it strikes me as odd to not update your own published data repository for up to four years – Why? This seems to not be in line with the authors key message of transparency and neither fits with the journal’s ethics of data sharing.

Minor

1. Introduction – authors suggest that there is lack of consistency in epidemiological stratification within the region, however, this paragraph is counterintuitive to their original premise of a ‘one-size-fits-all’ approach defined by the WHO will not be sufficient – consider revising this paragraph

2. Limitation of including averaged environmental covariates for a long temporal band (2010 – 2020) is worth acknowledging in the discussion. Using an averaged of the environment is not representation of the biological system of transmission.

3. The data included here from DHS is GPS displaced in urban areas by 2km and in rural up to 5km which can mean that prediction outputs are mis-specified. Whilst there isn’t much you can do I think it should be acknowledged in the discussion section about the DHS.

Figures

1. Figure 1 – figure 1A seems unnecessary and can be removed (though the legend can be consolidated with the PfPR legend). Figure 1C also doesn’t provide the reader much information and can be moved to the supplements allowing you to expand 1B as the points are difficult to see. Lastly, change the colour palette to something that is colour-blind friendly as I couldn’t distinguish the high from low.

2. Figure 2 – include legend in the figure itself rather than only in the caption. Figure 2B the y-axis title needs to be changed to Proper form.

3. Figure 3 – colour-blind friendly palettes

Reviewer #3: The manuscript “Plasmodium falciparum parasite prevalence in East Africa: updating data for malaria stratification” is an excellent article presenting methods for utilizing national malaria prevalence survey data for sub-national estimation of parasite prevalence. The spatial and statistical methods are sound and the authors’ integration of data across contiguous countries is fantastic, as malaria transmission is widely known to have great impacts in border regions due to divergence in national control programs. I have few major concerns and other minor critiques to be addressed.

Major critiques:

The justification for using large parasite prevalence surveys, conducted infrequently, for this work is a bit weak. Most countries now use dhis2 surveillance systems to monitor malaria incidence from all health facilities. Many have also integrated community health worker reporting into this system. This has a much higher spatial resolution and more detailed data. It is not publicly reported, but accessible through NMCPs. The primary research question is different, as these are clinical cases and do not represent parasite prevalence, but are generally used as a proxy for predicting malaria risk. The authors’ objective was to use the existing prevalence surveys to improve predictive risk maps, while acknowledging the limitations of these surveys. I would like to see a better justification for not using the real time passive surveillance systems to create predictive risk maps, particularly since these surveillance systems typically guide malaria control activities.

The authors also fall short on providing justification for the use of these models. It is unclear if this will lead to tools that can be used to inform control programs. This is of particular importance in areas of high prevalence in border regions. Will these methods be shared with NMCPs and how will it be used to guide the national programs? This is mentioned a bit in the discussion, but is not elaborated on and not presented in the introduction as a purpose of the study and potential public health impacts. The findings are great and would really be impactful if integrated into the programs.

Minor critiques:

Methods:

With the spatial scope there is incredible heterogeneity in malaria prevalence. The authors age standardized the prevalence measure to PfPR2-10. This is appropriate for high transmission settings where this age groups represents the highest risk group for transmission and identification. In lower transmission settings, this assumption does not hold, and infection reservoirs are primarily older adolescents and young adults. This could skew the results to make areas of low prevalence seem lower than they actually are. Justification for using this measure across varied transmission settings would be helpful.

More information about the RDT to microscopy conversion would be helpful. The use of these measures should also be addressed in the discussion. Both of these tools are diagnostic and not screening tools designed for measuring parasite prevalence. The sensitivity is quite poor in low transmission settings for this purpose and underestimate the prevalence. This could impact the areas that are presented as very low or low risk. Given the findings, there could be substantial misclassification just based on the data collection methods.

Results:

It would be helpful to present (in text) the proportion of the total population that is represented in this dataset.

In Figure 1 it is clear that most of the data comes from areas of higher population density. Was population density considered as a covariate in the model?

6. PLOS authors have the option to publish the peer review history of their article (what does this mean?). If published, this will include your full peer review and any attached files.

**Do you want your identity to be public for this peer review?** For information about this choice, including consent withdrawal, please see our Privacy Policy.

Reviewer #2: No

Reviewer #3: No

---

## [Decision Letter · Decision Letter 1]

15 Nov 2021

Plasmodium falciparum parasite prevalence in East Africa: updating data for malaria stratification.

PGPH-D-21-00103R1

Dear Dr. Alegana,

We're pleased to inform you that your manuscript has been judged scientifically suitable for publication and will be formally accepted for publication once it meets all outstanding technical requirements. 

Within one week, you'll receive an e-mail detailing the required amendments. When these have been addressed, you'll receive a formal acceptance letter and your manuscript will be scheduled for publication.

An invoice for payment will follow shortly after the formal acceptance. To ensure an efficient process, please log into Editorial Manager at https://www.editorialmanager.com/pgph/ click the 'Update My Information' link at the top of the page, and double check that your user information is up-to-date. If you have any billing related questions, please contact our Author Billing department directly at authorbilling@plos.org.

Kind regards,

Ruth Ashton, Ph.D.

Academic Editor

Additional Editor Comments (optional):

Thank you for your patience during this second round of reviews. Unfortunately only one of the original peer reviewers was able to provide feedback on the revised manuscript, but from their feedback and my own review I'm satisfied that all previous comments have been adequately addressed in this version. 

Reviewers' comments:

Reviewer's Responses to Questions

**Comments to the Author**

1. If the authors have adequately addressed your comments raised in a previous round of review and you feel that this manuscript is now acceptable for publication, you may indicate that here to bypass the “Comments to the Author” section, enter your conflict of interest statement in the “Confidential to Editor” section, and submit your "Accept" recommendation.

Reviewer #2: All comments have been addressed

2. Does this manuscript meet PLOS Global Public Health’s publication criteria? Is the manuscript technically sound, and do the data support the conclusions? The manuscript must describe methodologically and ethically rigorous research with conclusions that are appropriately drawn based on the data presented.

Reviewer #2: Yes

3. Has the statistical analysis been performed appropriately and rigorously?

Reviewer #2: Yes

4. Have the authors made all data underlying the findings in their manuscript fully available (please refer to the Data Availability Statement at the start of the manuscript PDF file)?

Reviewer #2: Yes

5. Is the manuscript presented in an intelligible fashion and written in standard English?

Reviewer #2: Yes

6. Review Comments to the Author

Reviewer #2: The paper is fundamental piece for malaria risk stratification. well done.

7. PLOS authors have the option to publish the peer review history of their article (what does this mean?). If published, this will include your full peer review and any attached files.

**Do you want your identity to be public for this peer review?** For information about this choice, including consent withdrawal, please see our Privacy Policy.

Reviewer #2: **Yes: **Punam Amratia
